# Real-World Experiences with Pazopanib in Patients with Advanced Soft Tissue and Bone Sarcoma in Northern California

**DOI:** 10.3390/medsci7030048

**Published:** 2019-03-18

**Authors:** Tiffany Seto, Mee-Na Song, Maily Trieu, Jeanette Yu, Manpreet Sidhu, Chi-Mei Liu, Danny Sam, Minggui Pan

**Affiliations:** 1Internal Medicine Residency Program, Kaiser Permanente, Santa Clara, CA, USA; Tiffany.seto@kp.org (T.S.); Mee-na.song@kp.org (M.-N.S.); Chi-mei.liu@kp.org (C.-M.L.); Danny.sam@kp.org (D.S.); 2Department of Drug Utilization, Kaiser Permanente, Oakland, CA 94612, USA; Maily.k.trieu@kp.org; 3Department of Oncology and Hematology, Kaiser Permanente, Oakland, CA 95051, USA; Jeanette.c.yu@kp.org (J.Y.); Manpreet.sidhu@kp.org (M.S.); 4Division of Research, Kaiser Permanente, Oakland, CA 94612, USA

**Keywords:** pazopanib, soft tissue sarcoma, bone sarcoma, real-world experiences, complete response, partial response, disease control rate, progression-free survival

## Abstract

**Background:** Pazopanib was approved for advanced soft tissue sarcoma as a second- or third-line therapy based on the clinical trial “Pazopanib for metastatic soft-tissue sarcoma” (PALETTE). We hypothesized that the real-world experiences may be significantly different from the clinical trial results. **Methods:** We analyzed the response pattern of patients with advanced soft tissue and bone sarcoma who received pazopanib treatment between 1 January 2011 and 31 October 2018 in Kaiser Permanente Northern California. **Results:** A total of 123 patients with 23 different histologic subtypes were assessable. One patient with low-grade fibromyxoid sarcoma obtained complete response (CR) after 2 months of treatment with pazopanib, 12 patients (9.7%) obtained partial response (PR), 34 patients (27.6%) had stable disease (SD), while 76 patients (61.8%) developed progressive disease (PD). The disease control rate (DCR) was 46.3% (CR + PR + SD). Among the 12 patients with PR, 3 had undifferentiated pleomorphic sarcoma (UPS), 4 had leiomyosarcoma (LMS), 2 had pleomorphic rhabdomyosarcoma, 1 had pleomorphic liposarcoma, 1 had dedifferentiated liposarcoma, and 1 had angiosarcoma. The median duration of response was 9 months. Two patients with Ewing’s sarcoma had SD for 6 and 13 months, and two patients with osteosarcoma had SD for 6 and 9 months. Among 65 patients assessed at 8 weeks, 9 had a response, and 10 had SD. Among 104 patients assessed at 12 weeks, 12 had a response, and 26 had SD. The median progression-free survival (PFS) was approximately 3 months for all 123 cases and for patients with UPS and LMS. **Conclusions:** Our cohort of patients with advanced soft tissue and bone sarcoma in Northern California treated with pazopanib had diverse histologic subtypes. The response rate (CR + PR) was higher than that of the PALETTE trial, while the DCR and the median PFS were significantly lower. The observation of PR in two patients with liposarcoma and durable SD in several patients with bone sarcoma indicates that pazopanib has activity in liposarcoma and bone sarcoma.

## 1. Introduction

Pazopanib was approved in 2012 as second- or third-line therapy for patients with metastatic soft tissue sarcoma (STS) on the basis of the “Pazopanib for metastatic soft-tissue sarcoma trial” (PALETTE), a randomized, placebo-controlled, double-blinded, phase III study that showed an approximately 3-month improvement of progression-free survival (PFS) when compared to the placebo [1]. The improvement of overall survival (OS) was less than 2 months compared to the placebo and was not statistically significant. Moreover, the PALETTE trial excluded adipocytic tumors, gastrointestinal stromal tumors (GIST), embryonal rhabdomyosarcomas, and bone sarcomas [1,2,3]. There have been case reports of response in patients with bone sarcoma treated with pazopanib [4,5,6]. Recently, a phase II study showed that regorafenib had activity in metastatic osteosarcoma [7].

Pazopanib is a tyrosine kinase inhibitor that targets multiple receptor tyrosine kinases mediating angiogenesis, including vascular endothelial growth factor receptor (VEGFR) and platelet-derived growth factor receptor (PDGFR) [8,9]. Several studies have shown expression of VEGF and VEGFR in soft tissue sarcomas [10,11,12]. The serum levels of VEGF were also found to be elevated in patients with metastatic soft tissue sarcomas [13,14]. Approved in 2009 for metastatic renal cell carcinoma (RCC) [9,15], pazopanib has been shown in phase II trials to have significant activity in other malignancies including thyroid carcinoma [16,17] and, more recently, in von Hippel–Lindau disease [18] and others. A phase II trial with 81 GIST patients treated with pazopanib as third-line therapy showed improved PFS [19], though in a separate phase II trial with 25 GIST patients treated after third-line therapy with pazopanib, the activity seemed limited [20].

Soft tissue and bone sarcomas are a group of very heterogeneous mesenchymal malignancies that show a diverse response to treatment. Some sarcomas are sensitive to chemotherapy and radiation, while others are minimally responsive to chemotherapy [21]. Subtypes of sarcoma also respond differently to various chemotherapy drugs and treatment regimens [22,23].

In this study, we examined the real-world experiences of pazopanib therapy in this diverse group of malignancies for clinical insight into real-world practice using our large electronic database. We identified 123 eligible cases from our Northern California Kaiser Permanente Cancer Registry between January 2011 and October 2018. We reviewed and analyzed the response pattern of 123 patients with different histologic subtypes of soft tissue and bone sarcoma and provide interesting results that can be helpful to the sarcoma clinical practice.

## 2. Materials and Methods

### 2.1. Study Design

This was a retrospective study of patients with advanced soft tissue and bone sarcoma who received pazopanib treatment between 1 January 2011 and 31 October 2018 in the Kaiser Permanente Northern California healthcare network (KPNC). The aim of this study was to determine the patterns of clinical benefit in the patients who received pazopanib treatment in the real-world community practice settings. This study was approved by the Institutional Review Board of Kaiser Permanente (protocol CN-17-2860).

### 2.2. Study Population

Kaiser Permanente Northern California is an integrated healthcare delivery system that serves a community-based population of 4.2 million with diverse ethnicity in Northern California. All eligible patients were identified using electronic medical and pharmacy records. All patients had histologically confirmed diagnosis, and pazopanib therapy was confirmed by reviewing electronic medical and pharmacy records. Each pathologic diagnosis was confirmed by a sarcoma pathologist in an academic institution including Stanford University, Mayo Clinic, Cleveland Clinic, etc.

### 2.3. Data Collection

The data on patient demographics and clinical information were abstracted from KPNC’s electronic medical records. The study endpoints were response to treatment and PFS. The response was assessed on the basis of the RECIST version 1.1 [24]. The disease control rate (DCR) was defined as the combined percentage of patients with complete response (CR), partial response (PR), and stable disease (SD). The PFS was defined as the period between the start of pazopanib treatment and the time of disease progression or death. Adverse events reported were based on the documentation by the treating oncologist. The median length of radiographic assessment after the initiation of pazopanib in our cohorts was approximately 10 weeks and at an interval of 8 to 12 weeks thereafter.

### 2.4. Statistical Analysis

We used descriptive analysis of CR, PR, SD, DCR, and PD for all cases and for histological subtypes with 10 or more patients. The PFS was assessed using the Kaplan–Meier plot. All endpoint analyses were conducted using the SAS software.

## 3. Results

From 1 January 2011 to 31 October 2018, a total of 133 patients treated with pazopanib were identified. We excluded 10 patients (3 patients with carcinosarcoma and 7 patients who took pazopanib for less than 4 days because of intolerance). The total number of patients for the final analysis was 123. The clinical characteristics of the 123 patients are presented in Table 1. The median age at which patients started pazopanib was 60 (ranging from 14 to 85) years. The cohort consisted of 52.8% women and 47.2% men who included 59% Caucasian, 20% Hispanic/Latino, 12% Asian/Pacific Islander, 8% African American, and 1% others. The median number of prior lines of treatment was three: 21% had one prior line of treatment, 23% had two, 35% had three, 21% had four or more lines of treatment. The median duration of treatment with pazopanib was approximately 3 months. Approximately 79% of patients started with 800 mg a day, 8% with 600 mg a day, 12% with 400 mg a day, and 1% with 200 mg a day. Approximately 90% of patients received prior anthracycline-based chemotherapy, and 70% received gemcitabine- and docetaxel-based regimen.

The responses for the 123 patients included 1 CR, 12 PR, 34 SD, and 76 PD (Table 2). The overall response rate was 10.6%, and the overall SD rate was 27.6%, with a DCR of 38.2% (Table 2). Sixty-five patients (52.8%) were assessable at 8 weeks, including 14 patients who progressed rapidly and symptomatically or died before 8 weeks without a radiographic assessment. Among these 65 patients, there were 9 responses (1 CR and 8 PR), 10 SD, and 46 PD. At 12 weeks, 104 patients (84.5%) were assessed, including 15 patients who progressed symptomatically or died before 12 weeks without radiographic assessment. Among these 104 patients, there were 12 responses (1 CR + 11 PR), 26 SD, and 66 PD. Among the 19 patients who had first radiographic assessment performed after week 12, there were1 PR and 8 SD (Table 2).

We determined the median PFS using Kaplan–Meier plot (Figure 1 and Figure 2). The median PFS was approximately 3 months for all 123 patients and for patients with undifferentiated pleomorphic sarcoma (UPS), leiomyosarcoma (LMS), as well as for the rest of the patients with other histology. Approximately 57% of all patients progressed within 3 months.

Of the 123 patients with 23 histologic subtypes (Table 3), one patient with low-grade fibromyxoid sarcoma of the lower back obtained CR after 2 months of treatment with pazopanib at 800 mg a day. This patient was diagnosed more than 30 years ago and had received numerous local therapies including surgery, radiosurgery, and external beam radiation for many relapses. After local therapy was no longer possible, the patient was referred for systemic therapy and obtained SD for 2 years and 5 months initially with tamoxifen treatment and then with anastrozole treatment (therapy change due to side effects of tamoxifen). Interestingly, this patient also obtained a very durable PR with gemcitabine and docetaxel chemotherapy after progression on anastrozole. Two months after pazopanib treatment, an MRI showed resolution of much of the nodular enhancement of the disease (Figure 3). Due to rapid tumor necrosis, the patient developed an abscess and open wound and had several debridement procedures performed. Interestingly, after all the debridement, there was no residual malignancy identified, indicating a pathologic CR. Among the 12 patients who obtained PR, one patient with metastatic leiomyosarcoma of the renal pelvis who relapsed more than 8 years after the initial surgery, obtained a PR after only 12 days of pazopanib at 600 mg a day (discontinued because of intolerance). This patient also obtained a very durable PR with gemcitabine/docetaxel chemotherapy as the prior line.

For the patient with low-grade fibromyxoid sarcoma who obtained CR, next-generation sequencing (NGS) using StrataNGS (Ann Arbor, MI, USA) was performed; however, no gene alteration was identified. The tumor mutation burden was 6 and there was no microsatellite instability (MSI). StrataNGS was also performed for the patient who obtained a PR after only 12 days of treatment with pazopanib (600 mg a day); no gene alteration other than RB1 deep deletion was identified. No tumor mutation burden or MSI was reported at the time of the test.

Among the histologic subtypes treated in our cohort (Table 3), the largest subtype was metastatic LMS (n = 40), followed by UPS (n = 22). Of the 40 patients with LMS, 4 (10%) had PR, and 9 (22.5%) had SD. For the 22 patients with UPS, 3 (13.6%) had PR, and 5 (22.7%) had SD. We observed PR in one of five patients with dedifferentiated liposarcoma, one of three patients with pleomorphic liposarcoma, two of three patients with pleomorphic rhabdomyosarcoma, one of four patients with angiosarcoma. In Figure 4, we show a PR observed in an 85-year-old patient with metastatic pleomorphic rhabdomyosarcoma after approximately 2 months of pazopanib. Both the primary tumor in the right shoulder and the metastatic masses in the lungs showed significant shrinkage that lasted for 9 months. This patient requested to try pazopanib as first-line because of his age and personal preference. In Figure 5, we show a durable PR lasting for 9 months observed in a 69-year-old patient with metastatic dedifferentiated liposarcoma after approximately 3 months of pazopanib treatment in the second-line setting.

The 34 patients with SD included 1 of 1 patient with angiomatoid histiocytoma, 1 of 4 patients with angiosarcoma, 9 of 40 patients with leiomyosarcoma, 5 of 22 patients with UPS, 4 of 7 patients with malignant peripheral nerve sheet tumor (MPNST), 1 of 1 patient with malignant phyllodes tumor, 1 of 3 patients with pleomorphic liposarcoma, and 4 of 7 patients with synovial sarcoma. Within the bone sarcoma subtypes, two of three patients with Ewing’s sarcoma had SD for 12 and 13 months respectively, two patients with osteosarcoma had SD for 6 and 9 months respectively, and three of eight patients with chondrosarcoma had SD.

## 4. Discussion

Soft tissue and bone sarcomas are a group of relatively uncommon and heterogenous mesenchymal malignancies with over 60 different histological subtypes, each with its own unique biological behavior and response to systemic therapy [25]. Ideally, treatment for advanced soft tissue and bone sarcomas should be tailored to the specific histologic subtype, but because of the tumor’s overall rarity, clinical trials have often grouped all sarcoma subtypes together. Consequentially, the results reported in clinical trials may not always translate into real-world clinical experiences.

For the 123 patients of our cohort here reported, the response rate was somewhat higher compared to that of the PALETTE trial (10.6% versus 6%); however, the DCR, or clinical benefit rate (CBR), and the SD rate were significantly lower (38.2% and 27.6% versus 73% and 67%). The median PFS also appeared significantly lower in our cohort versus that of the PALETTE trial (3 versus 4.6 months). One factor associated with lower DCR and SD in our cohort is likely related to the less frequent clinical and imaging assessments in the real-world practice during pazopanib treatment compared to the frequency of assessment during the clinical trial. The PALETTE trial specified clinical follow-up at 4, 8, and 12 weeks after the initiation of pazopanib and then at 8-week intervals [1]. The clinical assessment for the patients in our cohort was less frequent. This was supported by the fact that 9 out of the 19 patients who had first radiographic assessment after 12 weeks of pazopanib, had PR or SD, with a DCR of 47.3%. More patients might have been found to have SD had the assessment been more frequent. Another possible factor could be that the median number of previous lines of therapy was three for our cohort, compared to two in the PALETTE trial. In the PALETTE trial, the patients who had two to fourth lines of previous therapy had worse PFS compared to patients who had just first-line therapy [1]. The third factor could be related to poorer physical performance in our cohort. Though we do not have physical performance data in our cohort, more than one-third of our patients progressed within the first 2 months, and within these patients, one-third progressed symptomatically or died without an imaging study.

The patient with low-grade fibromyxoid sarcoma who obtained CR after only 2 months of treatment with pazopanib was particularly interesting. There was no gene alteration identified by a limited NGS panel. The other patient with LMS that obtained a PR after only 12 days of pazopanib was also interesting. Both patients had a prolonged PR to gemcitabine/docetaxel chemotherapy given as an every 14-day regimen as the first line of therapy (pazopanib was the second line). In the PALETTE trial, patients with grade I and II soft tissue sarcoma appeared to benefit more from pazopanib than the patients with grade III tumor [1]. Patients with locoregional disease, LMS, and synovial sarcoma histology appeared to derive more benefit, but the difference was not statistically significant [1]. We did not detect a significant difference with regard to the different histology in our cohort of patients.

The two patients with PR and one patient with SD among the eight patients with dedifferentiated or pleomorphic liposarcoma suggest that pazopanib can have activity in high-grade liposarcoma. Adipocytic tumors were excluded from the PALETTE trial because of the observation of low CBR (the progression-free rate (PFR) at 12 weeks was 26%) in a phase II EORTC trial [26]. Despite the small number of liposarcoma patients in our cohort, our results corroborate the findings from some of the more recent studies. A Japanese study that included 33 patients with liposarcoma showed 27% had SD at 8 weeks [27]. In another phase II study of 42 patients with advanced liposarcoma, the PFR at 12 weeks was 68.3% [28]. In both phase II trials, the most common histological subtype was dedifferentiated liposarcoma, followed by myxoid liposarcoma. Further analysis of the latter study showed the PFR at 12 weeks for dedifferentiated liposarcoma was 74.1%, and the PFR at 12 weeks for myxoid/round cell liposarcoma was 66.7%, but pleomorphic liposarcoma was not included in the analysis because of the low number of patients [28].

Our observation of durable SD in patients with Ewing’s sarcoma, osteosarcoma, and chondrosarcoma indicates that pazopanib retains significant activity in bone sarcomas. Previously, some case reports had shown PR in patients with extraosseous Ewing’s sarcoma treated with pazopanib [4,5,29]. Researchers from Denmark reported that three patients with advanced osteosarcoma obtained PR [6]. A study of 15 patients with advanced inoperable osteosarcoma treated with pazopanib reported that 3 patients obtained PR and 3 had SD [30]. Clinical trials evaluating pazopanib in combination with the inhibitors of mTOR, histone deacetylase (HDAC), MAPK, and ERBB4 pathways have been reported [31]. Other tyrosine kinase inhibitors (TKIs), including regorafenib, may have activity against bone sarcomas as well. A recent study reported by the French Sarcoma Group showed that patients with metastatic osteosarcoma treated with regorafenib versus placebo had prolonged PFS [32].

One of the limitations of our study is that it was a retrospective study including a very heterogenous mix of sarcoma subtypes over a period of several years, with most of the subtypes represented by less than 10 patients. The strength of our study is that it represents patients from a large institution with more than 4 million members of diverse ethnicities, yet, the treatment was provided in a community setting that is representative of the most common treatment settings in the United States.

## 5. Conclusions

Our study of 123 patients with advanced soft tissue and bone sarcoma treated with pazopanib showed a higher response rate compared to that of the PALETTE trial, but lower DCR, median PFS, and SD rate, suggesting that the real-world experiences with such treatment may be different from the reported clinical trial results. Understanding the real-world experiences can provide valuable insight for future clinical trial design and clinical practice.

## Figures and Tables

**Figure 1 medsci-07-00048-f001:**
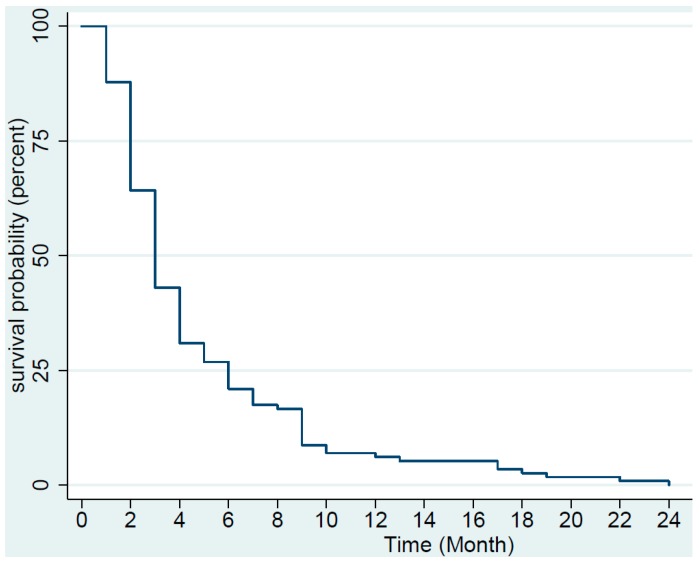
Kaplan–Meier plot of progression-free survival of the 123 patients with soft tissue and bone sarcoma treated with pazopanib. Two patients had not progressed and were censored (one with low-grade fibromyxoid sarcoma and one with synovial sarcoma).

**Figure 2 medsci-07-00048-f002:**
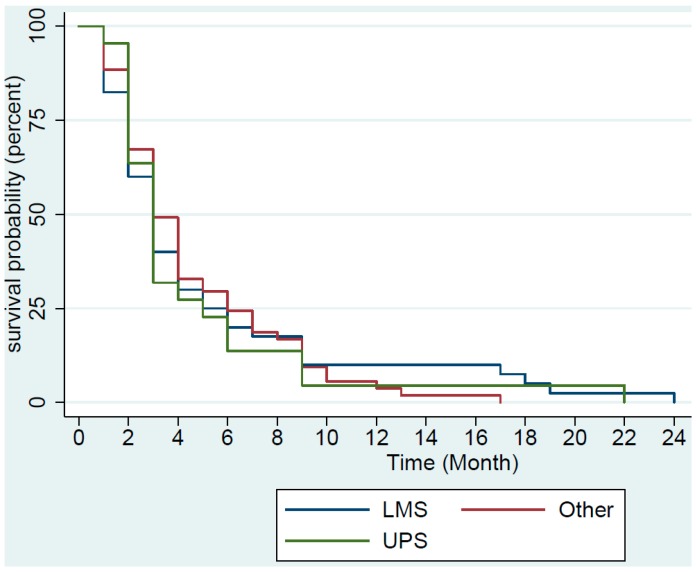
Kaplan–Meier plot of progression-free survival for the 40 patients with advanced leiomyosarcoma (LMS), 22 patients with undifferentiated pleomorphic sarcoma (UPS), and 61 patients with other types of soft tissue and bone sarcoma (Other). Two patients in the Other group had not progressed and were censored.

**Figure 3 medsci-07-00048-f003:**
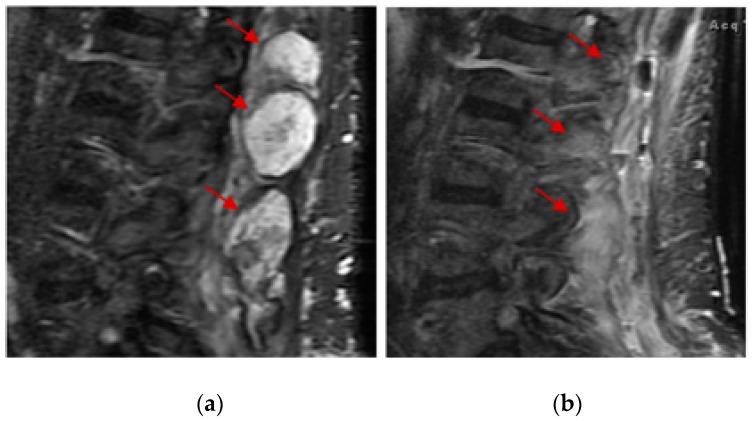
Complete response of a 57-year-old patient with low-grade fibromyxoid sarcoma of the lower back after 2 months of treatment with pazopanib. No malignancy was identified after debridement of the wound, indicating pathologic complete response. MRI of lumbar spine before (**a**) and after (**b**) pazopanib treatment. Red arrows indicate the location of the tumor masses.

**Figure 4 medsci-07-00048-f004:**
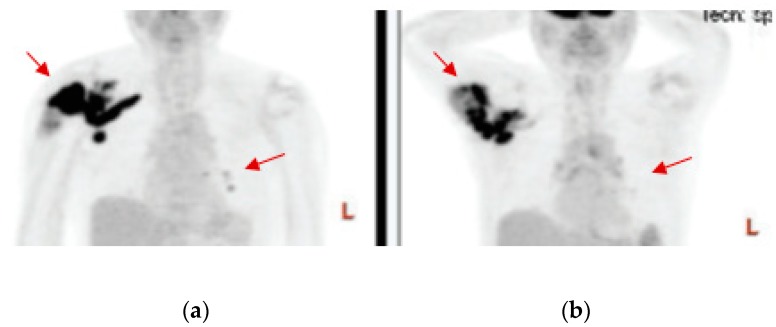
Partial response of an 85-year-old patient with pleomorphic rhabdomyosarcoma treated with pazopanib as first-line therapy. PET scans before (**a**) and at 8 weeks after (**b**) pazopanib show significant shrinkage of the primary tumor in the right shoulder as well as of the lung metastasis, as indicated by the red arrows.

**Figure 5 medsci-07-00048-f005:**
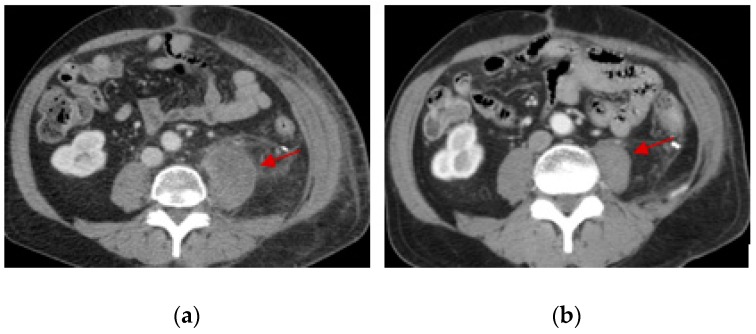
Partial response of a 69-year-old patient with dedifferentiated liposarcoma treated with pazopanib as second-line therapy. CT scans before (**a**) and 3 months after (**b**) pazopanib show significant shrinkage of the tumor that was attached to the left psoas muscle as indicated by the red arrow.

**Table 1 medsci-07-00048-t001:** Clinical characteristics of patients treated with pazopanib.

Median Age (Year)	Sex Distribution		Ethnicity				Median Lines of Prior Therapy
	Female	Male	White	Asian	Latino	African American	
60	52.8%	47.2%	60%	12%	19%	8%	3

**Table 2 medsci-07-00048-t002:** Response and disease control rate in all cases and in cases assessed at 8 and 12 weeks.

Response	All Cases (N = 123)	At 8 Weeks (N = 65)	At 12 Weeks (N = 104)	Cases Assessed after 12 Weeks (N = 19)
RR (PR + CR)	13 (12 +1) (10.6%)	9 (13.8%)	12 (11.5%)	1 (5.3%)
SD	34 (27.6%)	10 (15.4%)	26 (25%)	8 (42.1%)
PD	76 (61.8%)	46 (70.1%)	66 (63.5%)	10 (52.6%)
DCR (RR + SD)	47 (38.2%)	19 (29.2%)	38 (36.5%)	9 (47.4%)

RR, response rate; PR, partial response; CR, complete response; SD, stable disease; PD, progressive disease; DCR, disease control rate.

**Table 3 medsci-07-00048-t003:** Response to pazopanib based on histologic subtypes (the percentage of response is indicated for LMS, UPS, and all cases).

Histology	CR	PR	SD	PD	Total
Alveolar rhabdomyosarcoma				1	1
ASPS				1	1
Angiomatoid histiocytoma			1		1
Angiosarcoma		1	1	2	4
Chondrosarcoma			3	5	8
Chordoma				1	1
Dedifferentiated liposarcoma		1		4	5
Desmoplastic small round cell tumor				3	3
Epithelioid hemangioendothelioma				1	1
Ewing’s sarcoma			2	1	3
Endometrial stromal tumor				1	1
Hemangiopericytoma			1	1	2
Leiomyosarcoma		4 (10%)	9 (22.5%)	27 (67.5%)	40
Low grade fibromyxoid sarcoma	1				1
Malignant peripheral nerve sheath tumor			4	3	7
Malignant phyllodes			1		1
Myxoid liposarcoma				1	1
Osteosarcoma			2	4	6
Pleomorphic liposarcoma		1	1	1	3
Pleomorphic rhabdomyosarcoma		2		1	3
PeComa				1	1
Synovial sarcoma			4	3	7
UPS		3 (13.6%)	5 (22.7%)	14 (63.6%)	22
Total cases	N = 1 (0.8%)	N = 12 (9.7%)	N = 34 (27.6%)	N = 76 (61.8%)	N = 123 (100%)

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
