# Peer review of "Real-World Experiences with Pazopanib in Patients with Advanced Soft Tissue and Bone Sarcoma in Northern California"

_medsci, 2019, doi:10.3390/medsci7030048_

Round 1
Reviewer 1 Report
The manuscript reported the radiologic evaluation after administration of pazopanib in patients with advanced bone and soft tissue sarcoma (STS). The topic is interesting. But there are several concerns in the present manuscript.
1) Introduction. You mentioned the current circumstance of STS. But you don't introduce the bone sarcoma. Because you used pazopanib in patients with bone sarcoma, I suggest that you introduce the current circumstance of bone sarcoma.
2) Data collection. I wonder how you evaluate radiographic evaluations. Why PR? Why SD? RECIST criteria is easy to understand the radiographic effect. But you don't use it. How did you define PR, SD and PD? It should be necessary to show.
3) Please create Kaplan Meier curve of PFS.
4) Results. I would like to know the SD at 8 weeks or/and 12 weeks after the administration of pazopanib because some patients may be evaluated within 8 weeks. I think >8 weeks should be necesarry to determine the radiographic evaluation.
5) Please show the median PFS of some histologies which registered > 10 patients.
6) Please add the initial dose of pazopanib.
7) Please show the strategy for the patients with advanced bone sarcoma at your hospital. Pazopanib may not be indicated in patients with bone sarcoma.
Author Response
The manuscript reported the radiologic evaluation after administration of pazopanib in patients with advanced bone and soft tissue sarcoma (STS). The topic is interesting. But there are several concerns in the present manuscript.
Response--Thanks for the excellent suggestions, we have addressed all the concerns below.
1) Introduction. You mentioned the current circumstance of STS. But you don't introduce the bone sarcoma. Because you used pazopanib in patients with bone sarcoma, I suggest that you introduce the current circumstance of bone sarcoma.
Response--We have added references to the introduction as recommended.
2) Data collection. I wonder how you evaluate radiographic evaluations. Why PR? Why SD? RECIST criteria is easy to understand the radiographic effect. But you don't use it. How did you define PR, SD and PD? It should be necessary to show.
Response-- we have re-reviewed all the cases using RECIST criteria for responses.
3) Please create Kaplan Meier curve of PFS.
Response--We have created K-M curves for all the cases and also for the UPS and LMS cases (Figure 1 and 2).
4) Results. I would like to know the SD at 8 weeks or/and 12 weeks after the administration of pazopanib because some patients may be evaluated within 8 weeks. I think >8 weeks should be necesarry to determine the radiographic evaluation.
Response--we have also reviewed the response and SD at 8 and 12 weeks and reported the data in the result section.
5) Please show the median PFS of some histologies which registered > 10 patients.
Response--We have done so in Figure 2.
6) Please add the initial dose of pazopanib.
Response--we have added this data.
7) Please show the strategy for the patients with advanced bone sarcoma at your hospital. Pazopanib may not be indicated in patients with bone sarcoma.
Response--We have made comment on the Discussion section on this issue. It appears both regorafenib and pazopanib have good activity in bone sarcoma.
Reviewer 2 Report
See attached Word file: medsci-437369 Review

Author Response
Thank you for the excellent and thorough review. We have addressed all the concerns with substantial revision.
All of cases were reviewed by a sarcoma pathologist in an academic institution (Stanford, Mayo, Cleveland, etc.) we have added this information to the Materials and Methods section.
We have re-defined the eligibility and removed the patients who did not take or take for 3 days or less. We have added 6 more cases from June 2018 to October 2018. The total eligible cases are 123 cases now.
We have re-reviewed all the cases using RECIST criteria.
We have corrected the definition of DCR: Disease control rate (DCR) was defined as the percentage of patients with complete response (CR), partial response (PR) and stable disease (SD) combined.
We have corrected the errors in the Table. Thanks for pointing it out.
We have also corrected the error. There were 8 cases of chondrosarcoma, 3 had SD.
We have modified the table (Table 3) to make it easier to follow.
We have excluded the 3 cases with carcinosarcoma.
Minor issues:
we have also fixed all the minor issues, including PFS (median PFS), with Kaplan-Meier curves, and WHO reference as well as others. Thank you for the excellent review and recommendations that have substantially improved the quality of our manuscript.
Round 2
Reviewer 1 Report
The authors added the results and most part of those is satisfactory.
1) I think you should define the minimum duration of stable disease as SD evaluation. I think 8 weeks or 12 weeks is appropriate. If you do, on Table 2, you can show the response to the pazopanib easily. I think you should simply show the results of all cases (n=123). If you can't evaluate RECIST, those patients should be included in NE (not evaluated).
2) I don't agree with the reason for the difference of the response rate using RECIST evaluation in the discussion. If the clinical assessment in the present cohort was less frequent, those patients should be excluded from the evaluation (I mentioned above.). Instead, I think that you should discuss the difference of distribution of histology between the present study and PALETTE.
I believe you can improve those concerns.
Author Response
We thank the reviewer for the additional suggestions that have further improved the quality of our manuscript. We have addressed the questions below. I have highlighted the changes in the text with red.
In our manuscript, we have described the SD rate at 8-week and 12-week. We have also added the response data about the patients assessed after 12-week (see Table 2) and page 8 in the Result section, and page 11 at the Discussion section.
See page 12 in the Discussion section, we have added discussion on the histology subtypes in comparison to the PALETTE trial.